# Acute kidney injury associated with COVID-19: A retrospective cohort study

**Nitin V. Kolhe**[1,2]*, **Richard J. Fluck**[1,2], **Nicholas M. Selby**[1,2], **Maarten W. Taal**[1,2]

**1** Department of Nephrology, University Hospitals of Derby and Burton, Royal Derby Hospital, Derby, United Kingdom, **2** Centre for Kidney Research and Innovation, Division of Medical Sciences and Graduate Entry Medicine, School of Medicine, University of Nottingham, Derby, United Kingdom

* nitin.kolhe@nhs.net

## Abstract

### Background

Initial reports indicate a high incidence of acute kidney injury (AKI) in Coronavirus Disease 2019 (COVID-19), but more data are required to clarify if COVID-19 is an independent risk factor for AKI and how COVID-19–associated AKI may differ from AKI due to other causes. We therefore sought to study the relationship between COVID-19, AKI, and outcomes in a retrospective cohort of patients admitted to 2 acute hospitals in Derby, United Kingdom.

### Methods and findings

We extracted electronic data from 4,759 hospitalised patients who were tested for COVID-19 between 5 March 2020 and 12 May 2020. The data were linked to electronic patient records and laboratory information management systems. The primary outcome was AKI, and secondary outcomes included in-hospital mortality, need for ventilatory support, intensive care unit (ICU) admission, and length of stay. As compared to the COVID-19–negative group ($n = 3,374$), COVID-19 patients ($n = 1,161$) were older ($72.1 \pm 16.1$ versus $65.3 \pm 20.4$ years, $p < 0.001$), had a greater proportion of men (56.6% versus 44.9%, $p < 0.001$), greater proportion of Asian ethnicity (8.3% versus 4.0%, $p < 0.001$), and lower proportion of white ethnicity (75.5% versus 82.5%, $p < 0.001$). AKI developed in 304 (26.2%) COVID-19–positive patients (COVID-19 AKI) and 420 (12.4%) COVID-19–negative patients (AKI controls). COVID-19 patients aged 65 to 84 years (odds ratio [OR] 1.67, 95% confidence interval [CI] 1.11 to 2.50), needing mechanical ventilation (OR 8.74, 95% CI 5.27 to 14.77), having congestive cardiac failure (OR 1.72, 95% CI 1.18 to 2.50), chronic liver disease (OR 3.43, 95% CI 1.17 to 10.00), and chronic kidney disease (CKD) (OR 2.81, 95% CI 1.97 to 4.01) had higher odds for developing AKI. Mortality was higher in COVID-19 AKI versus COVID-19 patients without AKI (60.5% versus 27.4%, $p < 0.001$), and AKI was an independent predictor of mortality (OR 3.27, 95% CI 2.39 to 4.48). Compared with AKI controls, COVID-19 AKI was observed in a higher proportion of men (58.9% versus 51%, $p = 0.04$) and lower proportion with white ethnicity (74.7% versus 86.9%, $p = 0.003$); was more frequently associated with cerebrovascular disease (11.8% versus 6.0%, $p = 0.006$), chronic lung disease (28.0% versus 19.3%, $p = 0.007$), diabetes (24.7% versus 17.9%, $p = 0.03$), and CKD

**Data Availability Statement:** All relevant data are within the manuscript and its Supporting Information files.

**Funding:** The authors received no specific funding for this work.

**Competing interests:** I have read the journal's policy and the authors of this manuscript have the following competing interests: MWT is a member of the Editorial Board of PLOS Medicine NVK, RJF & NMS have declared that no competing interests exist.

**Abbreviations:** ACEI, angiotensin converting enzyme inhibitors; ACE2, angiotensin converting enzyme 2; AKI, acute kidney injury; ARB, angiotensin receptor blocker; CCF, congestive cardiac failure; CCMD, critical care minimum dataset; CCI, Charlson comorbidity index; CI, confidence interval; CKD, chronic kidney disease; COVID-19, Coronavirus Disease 2019; CRRT, continuous renal replacement therapy; CTD, connective tissue disorder; CVD, cerebrovascular disease; EPMA, electronic prescribing and medicine administration; EPR, electronic patient record; HR, hazard ratio; h-AKI, hospital-acquired AKI; ICD-10-CM, International Classification of Diseases, 10th Revision, Clinical Modification; ICNARC, Intensive Care National Audit and Research Centre; ICU, intensive care unit; IT, information technology; KDIGO, Kidney Disease Improving Global Outcomes; LIMS, laboratory information management system; MI, myocardial infarction; NHS, National Health Service; OR, odds ratio; PVD, peripheral vascular disease; RRT, renal replacement therapy; RT-PCR, real-time reverse transcriptase polymerase chain reaction; SARS-CoV-2, Severe Acute Respiratory Syndrome Coronavirus 2; SD, standard deviation; SLED, slow low efficiency dialysis; UHDB, University Hospitals of Derby and Burton.

(34.2% versus 20.0%, $p < 0.001$); and was more likely to be hospital acquired (61.2% versus 46.4%, $p < 0.001$). Mortality was higher in the COVID-19 AKI as compared to the control AKI group (60.5% versus 27.6%, $p < 0.001$). In multivariable analysis, AKI patients aged 65 to 84 years, (OR 3.08, 95% CI 1.77 to 5.35) and $\geq$85 years of age (OR 3.54, 95% CI 1.87 to 6.70), peak AKI stage 2 (OR 1.74, 95% CI 1.05 to 2.90), AKI stage 3 (OR 2.01, 95% CI 1.13 to 3.57), and COVID-19 (OR 3.80, 95% CI 2.62 to 5.51) had higher odds of death. Limitations of the study include retrospective design, lack of urinalysis data, and low ethnic diversity of the region.

## Conclusions

We observed a high incidence of AKI in patients with COVID-19 that was associated with a 3-fold higher odds of death than COVID-19 without AKI and a 4-fold higher odds of death than AKI due to other causes. These data indicate that patients with COVID-19 should be monitored for the development of AKI and measures taken to prevent this.

## Trial registration

ClinicalTrials.gov NCT04407156

## Author summary

### Why was this study done?

- Recent reports have suggested that some patients with Coronavirus Disease 2019 (COVID-19) develop acute kidney injury (AKI).

- There is a need to better understand risk factors for AKI in patients with COVID-19.

- It is also unclear if AKI in patients with COVID-19 differs from AKI due to other causes.

### What did the researchers do and find?

- In this study, we examined risk factors for AKI in patients with COVID-19 and also compared AKI in COVID-19 with AKI due to other causes.

- We found that males and patients of nonwhite ethnicity as well as those with comorbidities were at increased risk of developing AKI in COVID-19.

- AKI was associated with a 3-fold increase in mortality in COVID-19 patients.

- Patients with COVID-19 and AKI had higher mortality (60.5% versus 27.6%) than patients with AKI due to other causes, and COVID-19 was an independent predictor of mortality associated with an almost 4-fold odds of death.

**What do these findings mean?**

- COVID-19 frequently causes AKI, and when it does, it is associated with a higher mortality than COVID-19 without AKI or AKI due to other causes.

- Patients with COVID-19 should be monitored for early evidence of AKI so that preventive measures can be taken to avoid AKI.

## Introduction

The rapid progression of the global pandemic caused by the novel coronavirus, Severe Acute Respiratory Syndrome Coronavirus 2 (SARS-CoV-2), has resulted in an urgent need to understand the pathogenesis and variable clinical features of Coronavirus Disease 2019 (COVID-19). Lung involvement in the form of viral pneumonia, inflammatory infiltrates, and endothelial damage resulting in respiratory failure has been well documented and has been the focus of attention, but other organs including the kidneys are also affected in COVID-19 [1,2]. In the previous SARS epidemic, the reported incidence of acute kidney injury (AKI) was 6.7% with a high mortality of over 90% [3]. However, SARS-CoV-2 is a novel betacoronavirus belonging to the sarbecovirus subgenus of Coronaviridae family, and its effect on the kidneys is not yet fully understood. There have been reports of nephrology services being overwhelmed with new consults and the need for renal replacement therapy (RRT), which increased 18-fold during the COVID-19 pandemic [4]. In initial reports, AKI incidence in people with COVID-19 has ranged from 5% to 29% with substantial variation between centres, possibly due to differences in population demographics and risk factors for AKI [1,2,5–9]. Some reports on a small number of patients suggest that SARS-CoV-2 may have a specific effect on the kidneys, but it is not yet clear to what extent COVID-19 increases the risk of AKI or how AKI associated with COVID-19 may differ from AKI due to other causes [10,11].

In this study, we investigated the incidence and risk factors associated with AKI in patients admitted with COVID-19 and the impact of AKI on survival in 2 large acute hospitals in the UK. Further, to investigate possible unique features of AKI due to COVID-19, we studied patients with AKI from any cause and compared the clinical features and outcomes of those with and without COVID-19.

## Methods

### Study design and ethical approval

This was an investigator-initiated, multicentre, retrospective cohort study. The study protocol was assessed by the Research and Development Department of University Hospitals of Derby and Burton (UHDB) National Health Service (NHS) Trust and approved by the Health Research Authority and Wales Research Ethics Committee, and the study was registered in the National Library of Medicine website (www.clinicaltrials.gov) with registration number NCT04407156. The protocol is available as S1 Text. The research involved analysis of anonymised data routinely collected in the course of normal care and written informed consent was waived due to the nature of the study and pandemic nature of the disease. Data were analysed and interpreted by the authors who reviewed the manuscript and confirm the accuracy and completeness of the data and adherence to the protocol. The study was conducted according

to the principles expressed in the Declaration of Helsinki, and the results are reported according to the strengthening the reporting of observational studies in epidemiology (STROBE) guidelines (S1 Checklist).

## Participants and setting

This retrospective study was performed in UHDB NHS Foundation Trust. The Trust is comprised of 5 hospitals, of which 2 are acute hospitals with accident and emergency departments, serving a population of approximately 1.5 million people in Derbyshire and Staffordshire in the UK. We included all adult patients who were suspected of having COVID-19 infection and were admitted to the 2 acute hospitals between 5 March 2020 and 12 May 2020. During the study period, almost all elective admissions were cancelled. Clinical outcomes were collected until 13 May 2020, which was the final date of follow-up.

## Study design and procedures

All patients suspected of having COVID-19 infection and admitted to either of the 2 acute hospitals underwent nasal and pharyngeal swabbing. SARS-CoV-2 was detected using real-time reverse transcriptase polymerase chain reaction (RT-PCR), which was performed in the designated regional laboratory at Sheffield Teaching Hospitals NHS Foundation Trust. Antibody testing was not used to define COVID-19–positive cases. Patients were retested after 48 hours of a negative test, if there was high clinical suspicion of COVID-19 illness or if the swabbing process was judged to be inadequate. The criteria for COVID-19 testing in the UK included all patients who needed hospital admission because of clinical or radiological evidence of pneumonia or acute respiratory distress syndrome or influenza-like illness. Patients admitted with these presentations were tested regardless of travel history. We excluded the following patients: under the age of 18 years, not admitted to the hospital as per the above criteria, patients whose swab results were awaited, and those needing chronic maintenance haemodialysis or peritoneal dialysis. The microbiology database containing the results of nasal and pharyngeal swabs was linked to laboratory information management system (LIMS), hospital's electronic patient records (EPRs), electronic prescribing and medicine administration (EPMA) dataset, and critical care minimum dataset (CCMD). The hospital's LIMS has incorporated a nationally agreed (NHS England) AKI detection algorithm that generates automated real-time electronic alerts for AKI based on Kidney Disease Improving Global Outcomes (KDIGO) serum creatinine criteria with baseline creatinine defined as either the lowest creatinine available within 7 days or a median of serum creatinine values within 8 to 365 days [12,13]. The laboratory system then sends the test result using existing information technology (IT) connections to EPR. We extracted the first, the peak, and the last AKI stages along with corresponding dates of the test results. We also extracted intensive care unit (ICU) admission and discharge date and outcome disposition at discharge along with details of organ support from the CCMD. CCMD is used to collect daily data in critical care regarding reason for admission, organ support, length of stay, and outcome at discharge from critical care. During the study period, all RRTs were delivered as continuous renal replacement therapy (CRRT) except for 1 patient, who underwent slow low efficiency dialysis (SLED) treatment. We extracted data for patient demographics, comorbidities, and other diagnoses during hospital stay, which were based on codes from the International Classification of Diseases, 10th Revision, Clinical Modification (ICD-10-CM) and clinical outcomes that included length of stay, discharge method, and discharge destination. We extracted start and end date of any angiotensin converting enzyme inhibitors (ACEI) or angiotensin receptor blocker (ARB) prescription during inpatient stay. We included only the last admission for patients with multiple admissions during the observation period to

reduce selection bias. Patients who were not yet discharged from the hospital at the end of the study were recorded as alive. Charlson comorbidity index (CCI) was calculated from comorbidities that were extracted from the hospital EPR. Ethnicity was collected by the hospital self-reporting mechanism. Patients admitted during the study period who presented with or developed AKI but were SARS-CoV-2 PCR–negative were treated as controls.

## Study outcomes

The primary outcome was incident AKI. The secondary outcomes were in-hospital all-cause mortality, need for ventilatory support, ICU admission, and length of stay.

## Study definitions

Patients were suspected of having COVID-19 if the symptoms included fever greater than or equal to 37.8˚C and at least 1 of the following symptoms with acute onset: persistent cough (with or without sputum), hoarseness, nasal discharge or congestion, shortness of breath, sore throat, wheezing, and sneezing. The national criteria for screening did not include loss of taste or anosmia at the time of the study. AKI was identified using modified KDIGO definition of AKI as identified by NHS England's algorithm [14]. The algorithm compares the current measured serum creatinine from an individual patient against the baseline creatinine value defined as either the lowest in the last 7 days or a median of values from the preceding 8 to 365 days depending on availability of previous results stored in LIMS in real time. Urine output is not used in generating the AKI alerts in the AKI algorithm. Hospital-acquired AKI (h-AKI) was defined as AKI developing after 24 hours of hospital admission.

## Statistical analysis

The study followed the analysis plan as stated in the protocol (S1 Text) and can also be found at dx.doi.org/10.17504/protocols.io.bimskc6e. Descriptive statistical analysis was performed, and continuous variables are reported as mean with standard deviation (SD) and compared using a *t* test. Categorical variables are reported as proportion and percentages and were compared using chi-squared test or Fisher's exact test. Multiple imputation was not performed for missing data due to the very low proportion of missing data. If a patient's EPR or EPMA did not include information on comorbidity or use of ACEI or ARB, it was assumed that this information was absent.

We compared patients with confirmed COVID-19, referred to as "COVID-19 AKI," to those who did not develop AKI during the hospital stay, referred to as "COVID-19 controls." We also compared the characteristics of patients with COVID-19 who died versus those who survived. Further, we compared COVID-19 AKI patients with AKI patients who did not have COVID-19, referred to as "AKI controls" in the manuscript. Unadjusted associations between continuous and categorical variables in the groups were assessed by *t* test or the chi-squared test as appropriate. Comorbidities studied included myocardial infarction (MI), congestive cardiac failure (CCF), peripheral vascular disease (PVD), cerebrovascular disease (CVD), dementia, chronic lung disease, connective tissue disorder (CTD), diabetes with complications, paraplegia, chronic kidney disease (CKD), chronic liver disease, and cancer. Multivariable logistic regression analysis was used to investigate associations of baseline patient and clinical characteristics with outcomes defined as incident AKI or mortality. Age was not normally distributed and was log transformed. However, a Box–Tidwell procedure indicated a nonlinear relationship of log-transformed age with outcomes. We attempted several other transformations of age, but none was able to produce a linear relation to the logit of the dependent variable. We therefore adopted the approach of categorising age into 3 groups based on

the biological plausibility of COVID-19 affecting certain age-groups differently. Variables that were associated with outcomes in univariable analyses were included in the models. Results are presented as odds ratios (ORs) and 95% confidence intervals (CIs). To verify the robustness of our findings, we performed sensitivity analyses using CCI instead of individual comorbidities. In the first, we analysed the risk factors associated with AKI in COVID-19, and in the second, we assessed the factors associated with mortality in AKI patients. All tests were 2-tailed, and $p < 0.05$ was considered significant. Analysis was performed on IBM SPSS Statistics for Mac, Version 24.0 (IBM Corp, UK).

## Results

During the period from 5 March 2020 to 12 May 2020, 4,759 patients, who were tested for COVID-19, had 5,932 admissions to the UHDB. We excluded the following as per the exclusion criteria: 261 patients under the age of 18 years, 32 admissions in 21 patients on various forms of maintenance dialysis, and 4 duplicate records. There were 1,100 initial admissions with subsequent readmissions during the study period. In the final analysis, we included 4,535 admissions in 4,535 patients, of whom 1,161 were SARS-CoV-2 PCR–positive and 3,374 were SARS-CoV-2 PCR–negative. Data on comorbidities were missing in 14 patients, and ethnicity was missing in 7 patients. The overall incidence of AKI was 16%; AKI developed in 304 patients with COVID-19 (26.2%) and 420 patients without COVID-19 (12.4%) (Fig 1).

### Factors associated with AKI in COVID-19

Demographic, comorbidity, treatment, and outcome variables for patients with COVID-19 are shown in Table 1. As compared to COVID-19 controls, patients with COVID-19 AKI had a higher mean age (74.9 ± 12.8 years versus 71.1 ± 17.0 years, $p = 0.003$), but there was no difference in the age-groups. COVID-19 AKI patients had a higher prevalence of comorbidities: MI (13.8% versus 9.0%, $p = 0.021$), CCF (26.6% versus 14.7%, $p < 0.001$), CKD (34.2% versus 14.0%, $p < 0.001$), and chronic liver disease (3.0% versus 0.9%, $p = 0.024$). COVID-19 AKI patients were also more likely to require ICU admission (21.1% versus 3.7%, $p < 0.001$), mechanical ventilation (16.4% versus 3.6%, $p < 0.001$), evidenced a higher mortality (60.5% versus 27.4%, $p < 0.001$), and had a longer length of stay (9.3 ± 10.8 versus 7.1 ± 10.3 days, $p = 0.003$) than COVID-19 patients without AKI. AKI developed in 61.7% of COVID-19–positive patients who needed mechanical ventilation as compared to 23.5% of COVID-19–positive patients who did not need mechanical ventilation ($p < 0.001$).

Multivariable logistic regression analysis was performed using age, gender, ethnicity, residence in care home, use of ACEI/ARB, need for mechanical ventilation, and comorbidities as independent variables, selected on the basis of significant associations in univariable analyses. The analysis identified the following independent risk factors for AKI in patients with COVID-19: age 65 to 84 years (OR 1.67, 95% CI 1.11, 2.50), age ≥85 years (OR 1.66, 95% CI 1.01, 2.71), CCF (OR 1.72, 95% CI 1.18, 2.50), chronic liver disease (OR 3.43, 95% CI 1.17, 10.00), CKD (OR 2.81, 95% CI 1.97, 4.01), and needing mechanical ventilation (OR 8.74, 95% CI 5.17, 14.77) (Table 2).

### Factors associated with mortality in COVID-19

We compared demographic and clinical characteristics between survivors and non-survivors with COVID-19 (S1 Table). As compared to COVID-19 survivors, patients who died were older (68.9 ± 17.2 years versus 77.7 ± 12.0 years, $p < 0.001$), more likely to be male (53.1% versus 62.8%, $p = 0.02$) and care home residents (13.9% versus 19.1%, $p = 0.02$), more likely to require mechanical ventilation (5.4% versus 9.8%, $p = 0.006$), and had more comorbidities: MI

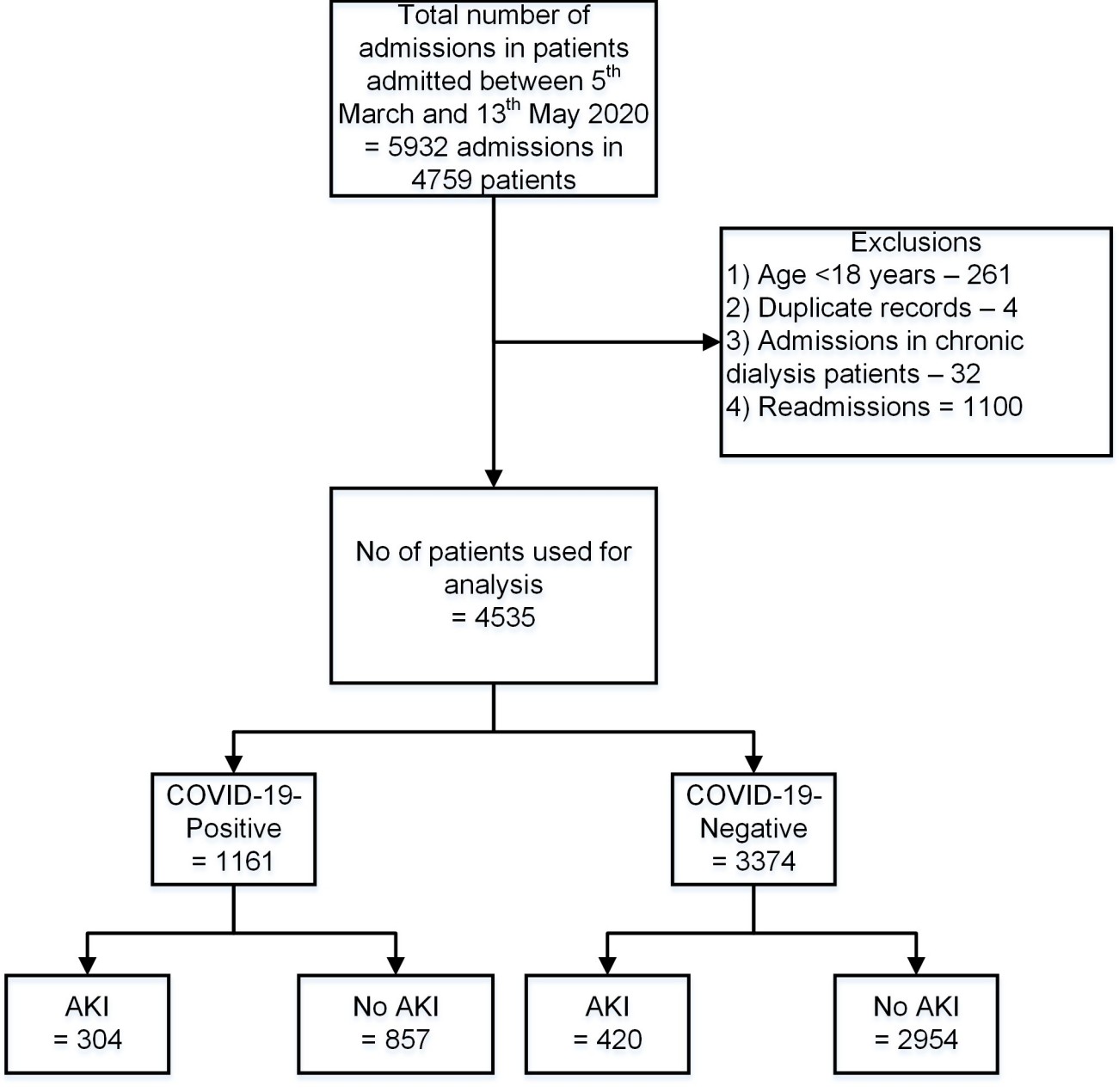

**Fig 1. Study flowchart showing the number of participants involved at each stage of the study.** AKI, acute kidney injury; COVID-19, Coronavirus Disease 2019.

(7.1% versus 15.8%, $p < 0.001$), CCF (12.7% versus 27.0%, $p < 0.001$), PVD (3.1% versus 10.3%, $p < 0.001$), dementia (9.3% versus 22.7%, $p < 0.001$), chronic lung disease (23.2% versus 33.2%, $p < 0.001$), diabetes with complications (19.1% versus 27.0%, $p = 0.002$), chronic liver disease (0.7% versus 2.9%, $p < 0.01$), CKD (12% versus 32.2%, $p < 0.001$), and cancer (5.4% versus 9.8%, $p = 0.006$). COVID-19 patients who died were also more likely to have developed AKI (16.2% versus 43.9%, $p < 0.001$). We performed multivariable logistic regression analysis to identify independent predictors of mortality in COVID-19. Higher age-group (age-group 65 to 84 years: OR 3.47, 95% CI 2.27, 5.3; age-group ≥85 years: OR 4.33, 95% CI 2.62,7.16), male sex

**Table 1. Patient characteristics in groups with and without AKI according to COVID-19 status.**

| | | COVID-19–positive | | COVID-19–negative | *p*-value | *p*-value |
|---|---|---|---|---|---|---|
| | | No AKI (A) | AKI (B) | AKI (C) | B vs. C | A vs. B |
| *DEMOGRAPHICS* | | | | | | |
| Number of patients | | 857 (73.8) | 304 (26.2) | 420 (12.4) | | |
| Age in years[†] | | 71.1 ± 17.0 | 74.9 ± 12.8 | 73.1 ± 16.7 | 0.125 | 0.003 |
| Age-group (years) | 18–64 | 253 (29.5) | 68 (22.4) | 102 (24.3) | 0.588 | 0.057 |
| | 65–84 | 405 (47.3) | 158 (52) | 202 (48.1) | | |
| | 85+ | 199 (23.2) | 78 (25.7) | 116 (27.6) | | |
| Gender | Male | 478 (55.8) | 179 (58.9) | 214 (51) | 0.041 | 0.381 |
| Ethnicity | White | 649 (75.7) | 227 (74.7) | 365 (86.9) | 0.003 | 0.443 |
| | Asian | 77 (9) | 19 (6.3) | 12 (2.9) | | |
| | Black | 13 (1.5) | 8 (2.6) | 6 (1.4) | | |
| | Not stated | 103 (12) | 43 (14.1) | 32 (7.6) | | |
| Care home residence | | 135 (15.8) | 48 (15.8) | 66 (15.7) | 1.000 | 1.000 |
| *COMORBIDITIES* | | | | | | |
| Myocardial infarction | | 77 (9) | 42 (13.8) | 40 (9.5) | 0.076 | 0.021 |
| Congestive cardiac failure | | 126 (14.7) | 81 (26.6) | 89 (21.2) | 0.092 | <0.001 |
| Peripheral vascular disease | | 44 (5.1) | 22 (7.2) | 21 (5) | 0.265 | 0.194 |
| Cerebrovascular disease | | 81 (9.5) | 36 (11.8) | 25 (6) | 0.006 | 0.267 |
| Dementia | | 119 (13.9) | 45 (14.8) | 49 (11.7) | 0.220 | 0.702 |
| Chronic lung disease | | 226 (26.4) | 85 (28) | 81 (19.3) | 0.007 | 0.598 |
| Connective tissue disorder | | 57 (6.7) | 17 (5.6) | 29 (6.9) | 0.538 | 0.586 |
| Diabetes with complications | | 180 (21) | 75 (24.7) | 75 (17.9) | 0.032 | 0.116 |
| Paraplegia | | 22 (2.6) | 7 (2.3) | 6 (1.4) | 0.407 | 1.000 |
| Chronic kidney disease | | 120 (14) | 104 (34.2) | 84 (20) | <0.001 | <0.001 |
| Chronic liver disease | | 8 (0.9) | 9 (3) | 18 (4.3) | 0.429 | 0.022 |
| Cancer | | 70 (8.2) | 32 (10.5) | 48 (11.4) | 0.721 | 0.238 |
| *AKI CHARACTERISTICS* | | | | | | |
| Peak AKI | Stage 1 | | 175 (57.6) | 251 (59.8) | 0.115 | |
| | Stage 2 | | 56 (18.4) | 93 (22.1) | | |
| | Stage 3 | | 73 (24) | 76 (18.1) | | |
| Hospital AKI | | | 186 (61.2) | 195 (46.4) | <0.001 | |
| AKI stage progression | | | 68 (22.4) | 59 (14) | 0.004 | |
| *TREATMENT* | | | | | | |
| ACEI or ARB use[¥] | | 124 (14.5) | 34 (11.2) | 66 (15.7) | 0.101 | 0.173 |
| Need for intensive care | | 32 (3.7) | 64 (21.1) | 43 (10.2) | <0.001 | <0.001 |
| Mechanical ventilation | | 31 (3.6) | 50 (16.4) | 32 (7.6) | <0.001 | <0.001 |
| Renal replacement therapy | | | 23 (7.6) | 9 (2.1) | <0.001 | |
| Renal support (days) | | | 2.4 ± 5.7 | 3.4 ± 7.1 | 0.502 | |
| *OUTCOMES* | | | | | | |
| Length of stay (days)[†] | | 7.1 ± 10.3 | 9.3 ± 10.8 | 8.9 ± 15.0 | 0.761 | 0.003 |
| Mortality | | 235 (27.4) | 184 (60.5) | 116 (27.6) | <0.001 | <0.001 |

¥ Angiotensin converting enzyme or angiotensin receptor blocker.

† Data are presented as mean ± standard deviation or number (percent).

AKI, acute kidney injury; COVID-19, Coronavirus Disease 2019.

(OR 1.4, 95% CI 1.05, 1.86), PVD (OR 2.44, 95% CI 1.35, 4.42), dementia (OR 2.27, 95% CI 1.49, 3.44), chronic liver disease (OR 4.37, 95% CI 1.27, 15.1), CKD (OR 1.7, 95% CI 1.18, 2.44),

**Table 2. Multivariable logistic regression to identify risk factors for AKI in COVID-19 disease.**

|  |  | Odds ratios | *p*-value |
|---|---|---|---|
| *DEMOGRAPHICS* |  |  |  |
| Age-group | 18–64 | 1 (Ref) |  |
|  | 65–84 | 1.67 (1.11, 2.5) | 0.013 |
|  | 85+ | 1.66 (1.01, 2.71) | 0.045 |
| Gender | Male | 0.99 (0.74, 1.33) | 0.955 |
| Ethnicity | White | 1 (Ref) |  |
|  | Asian | 0.89 (0.49, 1.6) | 0.687 |
|  | Black | 2.29 (0.88, 6.02) | 0.091 |
|  | Not stated | 1.14 (0.74, 1.77) | 0.541 |
| Care home residence |  | 1.01 (0.66, 1.55) | 0.962 |
| *COMORBIDITIES* |  |  |  |
| Myocardial infarction |  | 1.18 (0.74, 1.86) | 0.486 |
| Congestive cardiac failure |  | 1.72 (1.18, 2.5) | <0.001 |
| Peripheral vascular disease |  | 1.05 (0.58, 1.9) | 0.860 |
| Cerebrovascular disease |  | 1.04 (0.63, 1.73) | 0.865 |
| Dementia |  | 0.87 (0.56, 1.36) | 0.539 |
| Chronic lung disease |  | 0.95 (0.68, 1.32) | 0.756 |
| Connective tissue disorder |  | 0.69 (0.38, 1.28) | 0.242 |
| Diabetes with complications |  | 1.06 (0.75, 1.5) | 0.748 |
| Paraplegia |  | 0.69 (0.26, 1.87) | 0.470 |
| Chronic kidney disease |  | 2.81 (1.97, 4.01) | <0.001 |
| Chronic liver disease |  | 3.43 (1.17, 10) | 0.024 |
| Cancer |  | 1.37 (0.85, 2.22) | 0.198 |
| *TREATMENT* |  |  |  |
| ACEI or ARB use[¥] |  | 0.69 (0.45, 1.08) | 0.108 |
| Mechanical ventilation |  | 8.74 (5.17, 14.77) | <0.001 |

¥ Angiotensin converting enzyme or angiotensin receptor blocker.

The model was statistically significant, $\chi 2(4) = 148.1$, $p < 0.001$, and Hosmer–Lemeshow test was not significant, $p = 0.78$.

AKI, acute kidney injury; COVID-19, Coronavirus Disease 2019.

cancer (OR 3.02, 95% CI 1.88, 4.85), need for mechanical ventilation (OR 3.02, 95% CI 1.87, 5.73), and development of AKI (OR 3.27, 95% CI 2.39, 5.73) remained independent predictors of mortality, but the association of use of ACEI or ARB in COVID-19 with all-cause mortality was no longer statistically significant (OR 0.79, 95% CI 0.52, 1.19) (Table 3).

## Comparison of AKI in patients with COVID-19 and non-COVID-19 disease

We compared characteristics of the 304 patients with COVID-19 AKI with the group of 420 patients who developed AKI but were COVID-19–negative, referred to as "AKI controls," admitted during the same period (Table 1). There was no difference in mean age or the age-groups between the 2 groups. As compared to AKI controls, COVID-19 AKI evidenced a greater proportion of men (58.9% versus 51.0%, $p = 0.041$), lower proportion with white ethnicity (74.7% versus 86.9%, $p = 0.003$), more h-AKI (61.2% versus 46.4%, $p < 0.001$), greater proportion needing ICU (21.1% versus 10.2%, $p < 0.001$), and mechanical ventilation (16.4% versus 7.6%, $p < 0.001$). COVID-19 AKI patients also evidenced a greater proportion with

**Table 3. Multivariable logistic regression to identify risk factors for mortality in COVID-19 disease.**

| | | Odds ratios | p-value |
|---|---|---|---|
| *DEMOGRAPHICS* | | | |
| Age-group COVID-19 | 18–64 | 1 (Ref) | |
| | 65–84 | 3.47 (2.27, 5.3) | <0.001 |
| | 85+ | 4.33 (2.62, 7.16) | <0.001 |
| Gender | Male | 1.39 (1.05, 1.86) | 0.020 |
| Ethnicity | White | 1 (Ref) | |
| | Asian | 1.20 (0.68, 2.14) | 0.527 |
| | Black | 0.97 (0.31, 3.05) | 0.965 |
| | Mixed | 2.59 (0.39, 17.19) | 0.323 |
| | Others | 2.01 (0.6, 6.75) | 0.261 |
| | Not stated | 1.02 (0.66, 1.58) | 0.940 |
| Care home residence | | 0.86 (0.58, 1.3) | 0.480 |
| *COMORBIDITIES* | | | |
| Myocardial infarction | | 1.47 (0.93, 2.31) | 0.096 |
| Congestive cardiac failure | | 1.38 (0.95, 1.99) | 0.090 |
| Peripheral vascular disease | | 2.44 (1.35, 4.42) | <0.001 |
| Cerebrovascular disease | | 0.75 (0.45, 1.25) | 0.271 |
| Dementia | | 2.27 (1.49, 3.44) | <0.001 |
| Chronic lung disease | | 1.27 (0.92, 1.74) | 0.146 |
| Connective tissue disorder | | 1.21 (0.71, 2.07) | 0.487 |
| Diabetes with complications | | 1.15 (0.82, 1.61) | 0.411 |
| Paraplegia | | 1.35 (0.55, 3.34) | 0.513 |
| Chronic kidney disease | | 1.69 (1.18, 2.44) | <0.001 |
| Chronic liver disease | | 4.37 (1.27, 15.1) | 0.020 |
| Cancer | | 3.02 (1.88, 4.85) | <0.001 |
| *TREATMENT* | | | |
| ACEI or ARB use[¥] | | 0.79 (0.52, 1.19) | 0.252 |
| Mechanical ventilation | | 3.28 (1.87, 5.73) | <0.001 |
| *OUTCOME* | | | |
| AKI | | 3.27 (2.39, 4.48) | <0.001 |

[¥] Angiotensin converting enzyme or angiotensin receptor blocker.

The model was statistically significant, $\chi2(4) = 292.2$, $p < 0.001$, and Hosmer–Lemeshow test was not significant, $p = 0.50$.

AKI, acute kidney injury; COVID-19, Coronavirus Disease 2019.

comorbidities: CVD (11.8% versus 6.0%, $p = 0.006$), chronic lung disease (28.0% versus 19.3%, $p = 0.007$), diabetes with complication (24.7% versus 17.9%, $p = 0.032$), and CKD (34.2% versus 20.0%, $p < 0.001$). COVID-19 AKI patients were more likely than AKI controls to progress to higher AKI stages (22.4% versus 14.0%, $p = 0.004$) and need RRT (7.6% versus 2.1%, $p < 0.001$), but there was no difference in peak AKI stages or the length of stay between the 2 groups.

## Factors associated with mortality in patients with AKI

Overall, we observed 300 deaths in patients with AKI (41.4%), 184 (60.5%) in the COVID-19 AKI group versus 116 (27.6%) in AKI controls ($p < 0.001$). In univariable analyses, other factors associated with mortality in patients with AKI included age, gender, comorbidities, use of ACEI/ARB, ICU admission, mechanical ventilation, need for RRT, peak AKI stages, h-AKI,

and progression of AKI stages (S2 Table). In multivariable logistic regression analysis, AKI patients in the 65 to 84 years age-group, (OR 3.08, 95% CI 1.77, 5.35) and ≥85 years (OR 3.54, 95% CI 1.87, 6.70), peak AKI stage 2 (OR 1.74, 95% CI 1.05, 2.90) and stage 3 (OR 2.01, 95% CI 1.13, 3.57), and progression of AKI to higher stages (OR 1.85, 95% CI 1.04, 3.31) had higher odds of death (Table 4). Amongst comorbidities, CKD was not associated with increased odds of death (OR 1.31, 95% CI 0.83, 2.08), but dementia (OR 2.17, 95% CI 1.19, 3.97), paraplegia

**Table 4. Multivariable logistic regression to identify risk factors for mortality in patients with AKI.**

| | | Odds ratios | *p*-value |
|---|---|---|---|
| *DEMOGRAPHICS* | | | |
| Age-group | 18–64 | 1 (Ref) | |
| | 65–84 | 3.08 (1.77, 5.35) | <0.001 |
| | 85+ | 3.54 (1.87, 6.7) | <0.001 |
| Gender | Male | 1.27 (0.88, 1.82) | 0.201 |
| Ethnicity | White | 1 (Ref) | |
| | Asian | 1.49 (0.62, 3.58) | 0.375 |
| | Black | 2.77 (0.75, 10.24) | 0.128 |
| | Not stated | 1.52 (0.85, 2.73) | 0.157 |
| Care home residence | | 0.79 (0.45, 1.4) | 0.432 |
| *COMORBIDITIES* | | | |
| Myocardial infarction | | 1.29 (0.69, 2.38) | 0.432 |
| Congestive cardiac failure | | 2.61 (1.64, 4.15) | <0.001 |
| Peripheral vascular disease | | 1.14 (0.54, 2.44) | 0.728 |
| Cerebrovascular disease | | 0.62 (0.32, 1.23) | 0.172 |
| Dementia | | 2.17 (1.19, 3.97) | 0.012 |
| Chronic lung disease | | 1.39 (0.91, 2.16) | 0.131 |
| Connective tissue disorder | | 1.44 (0.69, 2.97) | 0.330 |
| Diabetes with complications | | 0.89 (0.56, 1.39) | 0.597 |
| Paraplegia | | 9.95 (1.98, 49.94) | 0.005 |
| Chronic kidney disease | | 1.31 (0.83, 2.08) | 0.250 |
| Chronic liver disease | | 4.65 (1.72, 12.58) | <0.001 |
| Cancer | | 2.49 (1.43, 4.38) | <0.001 |
| *COVID-19 STATUS* | | | |
| COVID-19–positive | | 3.79 (2.62, 5.51) | <0.001 |
| *AKI CHARACTERISTICS* | | | |
| Peak AKI | Stage 1 | 1 (Ref) | |
| | Stage 2 | 1.74 (1.05, 2.9) | 0.032 |
| | Stage 3 | 2.01 (1.13, 3.57) | 0.017 |
| Hospital AKI | | 1.26 (0.86, 1.86) | 0.238 |
| AKI stage progression | | 1.85 (1.04, 3.31) | 0.037 |
| Renal replacement therapy | | 1.61 (0.62, 4.23) | 0.331 |
| *TREATMENT* | | | |
| ACEI or ARB use¥ | | 0.48 (0.27, 0.85) | 0.012 |
| Mechanical ventilation | | 1.52 (0.81, 2.87) | 0.194 |

¥ Angiotensin converting enzyme or angiotensin receptor blocker.

The model was statistically significant, $\chi2(4) = 228.0$, $p < 0.001$, and Hosmer–Lemeshow test was not significant, $p = 0.59$.

AKI, acute kidney injury; COVID-19, Coronavirus Disease 2019.

(OR 9.95, 95% CI 1.98, 49.94), chronic liver disease (OR 4.64, 95% CI 1.72, 12.58), and cancer (OR 2.50, 95% CI 1.43, 4.38) were associated with higher odds of death. AKI patients who had COVID-19 had higher odds of death than COVID-19–negative patients with AKI (OR 3.8, 95% CI 2.62, 5.51). Patients who were received ACEI/ARB had lower odds of death (OR 0.48, 95% CI 0.27, 0.85).

Sensitivity analysis using CCI instead of individual comorbidities showed similar findings. In both analyses, increasing CCI was an independent risk factor for higher mortality (S3 and S4 Tables).

## Discussion

In this retrospective, multicentre study, we found a high incidence of AKI affecting more than a quarter of hospitalised patients with COVID-19. Independent predictors of AKI included age, CCF, CKD, and chronic liver disease along with mechanical ventilation. The impact of AKI on outcomes in the context of COVID-19 was demonstrated by the finding that AKI was independently associated with 3-fold higher odds of death. Furthermore, by comparing AKI in patients with and without COVID-19, we observed that AKI associated with COVID-19 was independently associated with an almost 4-fold higher odds of death than AKI associated with other acute illnesses.

### Risk factors for AKI in COVID-19

We found the incidence of AKI in COVID-19 patients was more than double the incidence of AKI in non-COVID-19 patients. The incidence of AKI in COVID-19 in our study is similar to that reported from the United States of America (22.2% to 36%) but much higher than that reported from China (5.1% to 10.5%) [1,2,6,7,15,16]. The variation in the incidence of AKI in COVID-19 in different countries or regions may be explained in part by variable inclusion criteria (intensive care and all hospital admissions) and also the varying demographic characteristics and comorbidities of study populations. Our finding of greater age as a risk factor for AKI is in keeping with 2 previous studies of AKI in COVID-19, though our study population was older (mean age: 73 years) than in the study from the US (mean age: 69 years) and much older than the Chinese cohort (mean age: 63 years) [5,15]. Though a higher proportion of men and patients of Asian ethnicity with COVID-19 had AKI, we did not find that male sex or ethnicity were independent risk factors for AKI. In contrast, studies from the US have reported men and black ethnic groups at increased risk for developing AKI in COVID-19 [15,17]. Intensive Care National Audit and Research Centre (ICNARC) has also reported lower proportion of patients from Asian and black ethnicity discharged alive from ICU [18]. This apparent discrepancy may be explained in part by the very low proportion of people of black ethnicity in the local population resulting in reduced statistical power to study the impact of ethnicity.

We also confirmed that several comorbidities and the need for mechanical ventilation were independent risk factors for AKI. Large variation has been reported in the prevalence of comorbidities in patients with COVID-19 and AKI. For example, we observed that diabetes was present in 25% of patients with COVID-19 and AKI, whereas the proportion with diabetes in studies from the US has been reported as 41% to 47% and from China as 14% [5,6,15]. We did not identify diabetes as risk factor for AKI or mortality in contrast to a study from the US [15]. This may be because of the dominant effects of other comorbidities on mortality and warrants further investigation. We found prior CCF as well as CKD and liver disease to be strong predictors of AKI in COVID-19, in keeping with data from another study which found that the incidence of AKI was higher in patients presenting with baseline creatine above normal (12 out of 101, 11.9% versus 24 out of 600, 4%) [5].

## Risk factors for mortality in COVID-19

Our findings confirmed that AKI is a strong independent risk factor for mortality in patients with COVID-19, associated with a 3-fold increase in odds of in-hospital death. The impact of AKI on mortality in COVID-19 has been reported in only 1 study to date [5]. Amongst 701 patients with COVID-19, AKI stages 2 and 3 were associated with increasing hazard ratios for in-hospital death (stage 2: hazard ratio [HR] 3.53, 95% CI 1.50, 8.27 and stage 3: HR 4.72, 95% CI 2.55, 8.75). We have also confirmed a strong association between older age and male sex with mortality in COVID-19. Two large studies have reported that men and people from Asian and black ethnicity were at markedly increased risk of in-hospital death from COVID-19 [19,20]. Our findings confirm that previous CKD, dementia, chronic liver disease, and cancer are independent predictors of mortality [20]. Previous studies have indicated that SARS--CoV-2 uses angiotensin converting enzyme 2 (ACE2) as a cell entry receptor, prompting some investigators to suggest that treatment with ACEI or ARB may increase the risk of severe complications associated with COVID-19 [21]. This has been questioned by others, and we found no increase in mortality associated with ACEI or ARB treatment [22].

## AKI in patients with COVID-19 versus non-COVID-19 disease

A unique aspect of this study is that we were able to compare AKI associated with COVID-19 with AKI due to other causes. We identified several differences between these groups including higher proportion of men, patients from Asian and black ethnicity, and h-AKI. A greater proportion of patients with COVID-19 AKI evidenced AKI stage progression and needed mechanical ventilation and intensive care. We found that 7.6% of COVID-19 AKI patients needed RRT in contrast to 2.1% in AKI controls. The need for RRT has ranged from 0.8% to 9% in China as compared to the US, where it has ranged from 14.3% to 55% [1,6,15,16]. The aetiology of AKI in COVID-19 seems to be different than usual AKI in hospitalised patients. It tends to occur late when patients are critically unwell, need mechanical ventilation, and vasopressor support. However, there is also emerging evidence that kidneys are affected early in COVID-19. Proteinuria and haematuria have been reported in 44% and 26.7% on admission, respectively [5,23]. This suggests that there are a number of different causes of AKI in COVID-19, and some mechanisms by which COVID-19 affects kidneys remain unclear. Experimental studies in human kidney proximal tubular epithelial cells have shown persistent infection with SARS-CoV-2 [24]. Further, transmission electron microscopy of kidneys in patients who died of COVID-19 have demonstrated virus particles in cytoplasm of proximal tubular cells, podocytes, and also in distal tubules [11]. These findings may suggest possible mechanisms of proteinuria and AKI in COVID-19 patients. Recently, collapsing glomerulopathy has been reported [10]. In spite of haematuria and proteinuria occurring early in the course of illness, we found that AKI in COVID-19 tends to develop later than non-COVID-19 AKI. This is in keeping with another study which reported that 62.7% of AKI developed after 24 hours, which in our study was defined as h-AKI. This may suggest that the causes of AKI in COVID-19 may not be predominantly classical prerenal causes; direct effects of SARS-CoV-2 on the kidneys and the inflammatory effect of high cytokine levels (cytokine storm) may be additional relevant factors. The practical implication of this observation is that patients with COVID-19 who do not have AKI on admission should have daily monitoring to detect h-AKI [25].

## Risk factors for mortality in AKI

In all patients with AKI, we found increasing age, higher AKI stages, and AKI stage progression were associated with increasing odds of death. Increasing AKI stage has also been found

to be associated with increased mortality in another study of COVID-19 with AKI. The authors found mortality of 33.7% (34 out of 101) and had higher hazards of death (HR of 4.72 with AKI stage 3) [5]. In our study, the in-hospital mortality in AKI patients with COVID-19 was 60.5% as compared to 27.6% in AKI controls. In other studies of AKI in COVID-19, in-hospital mortality ranged from 34.8% to 72% in the US and 16.1% to 86.4% in China [5,6,15,17,23].

Importantly, we found that COVID-19 was an independent risk factor for mortality in patients with AKI, associated with an almost 4-fold higher odds of in-hospital mortality. In the analysis of all patients with AKI, use of an ACEI or ARB was associated with lower odds of death, but as this was an observational study causality cannot be inferred. We did not find any association between need for RRT and mortality in COVID-19 AKI; this may be due to a low proportion of patients needing RRT in this study.

The retrospective and database nature of the study comes with some limitations. We did not have access to urinalysis results, and hence it is difficult to deduce if COVID-19 affects kidneys earlier than the biochemical changes, which appear later. We were unable to obtain the cause of AKI or the long-term outcomes due to a relatively short observation period, and this may have generated survivor bias. Further studies with longer observation periods are required to understand the long-term impact of COVID-19 on the kidneys. The East Midlands has a low proportion of people of black and Asian ethnicity, and this study may have lacked statistical power to investigate the effect of COVID-19 in ethnic minority groups. In addition, we did not have other laboratory investigations that may have helped to understand the severity of COVID-19. Finally, we have tried to minimise the magnitude of unmeasured confounders by having a COVID-19–negative AKI control, but, in an observational study, this can never be completely eliminated.

In conclusion, we found high incidence of AKI in patients with COVID-19 that was independently associated with greater age and comorbidities. AKI was associated with a 3-fold higher odds of death in patients with COVID-19, and patients with COVID-19 and AKI were at 4-fold higher odds of in-hospital death than those with AKI due to other causes. These data provide robust evidence to support that patients with COVID-19 should be closely monitored for the development of AKI and measures taken to prevent this, though further studies are required to determine the most effective clinical approach.

## Supporting information

**S1 Text. Study Protocol v1.2.**
(DOCX)

**S1 Checklist. STROBE checklist.**
(DOCX)

**S1 Table. Baseline characteristics between survivors and non-survivors with COVID-19.**
(DOCX)

**S2 Table. Univariate analysis of risk factors in survivors versus non-survivors in patients who developed AKI.**
(DOCX)

**S3 Table. Predictors of mortality in COVID-19 disease including Charlson comorbidity index.**
(DOCX)

**S4 Table. Predictors of mortality in AKI including Charlson comorbidity index.**
(DOCX)

## Acknowledgments

The authors gratefully acknowledge the information and technology support provided by Simon Randle from the University Hospitals of Derby and Burton.

## Author Contributions

**Conceptualization:** Nitin V. Kolhe, Richard J. Fluck, Nicholas M. Selby, Maarten W. Taal.

**Data curation:** Nitin V. Kolhe.

**Formal analysis:** Nitin V. Kolhe.

**Methodology:** Nitin V. Kolhe.

**Resources:** Nitin V. Kolhe.

**Supervision:** Maarten W. Taal.

**Validation:** Maarten W. Taal.

**Writing – original draft:** Nitin V. Kolhe.

**Writing – review & editing:** Nitin V. Kolhe, Nicholas M. Selby, Maarten W. Taal.

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
