## [Editor Report · Decision Letter 0]

22 Jun 2020

Dear Dr Kolhe, 

Thank you for submitting your manuscript entitled "Acute Kidney Injury associated with COVID-19: A retrospective cohort study" for consideration by PLOS Medicine.

Your manuscript has now been evaluated by the PLOS Medicine editorial staff [as well as by an academic editor with relevant expertise] and I am writing to let you know that we would like to send your submission out for external peer review.

Kind regards,

Adya Misra, PhD,

Senior Editor

PLOS Medicine

---

## [Decision Letter · Decision Letter 1]

10 Jul 2020

Dear Dr. Kolhe,

Thank you very much for submitting your manuscript "Acute Kidney Injury associated with COVID-19: A retrospective cohort study" (PMEDICINE-D-20-02816R1) for consideration at PLOS Medicine. 

[LINK]

In light of these reviews, I am afraid that we will not be able to accept the manuscript for publication in the journal in its current form, but we would like to consider a revised version that addresses the reviewers' and editors' comments. Obviously we cannot make any decision about publication until we have seen the revised manuscript and your response, and we plan to seek re-review by one or more of the reviewers. 

We expect to receive your revised manuscript by Jul 31 2020 11:59PM. Please email us (plosmedicine@plos.org) if you have any questions or concerns.

We look forward to receiving your revised manuscript. 

Sincerely,

Adya Misra, PhD

Senior Editor 

PLOS Medicine

plosmedicine.org

Comments from AE

The manuscript focuses on a topic of current interest, namely the development of acute kidney injury (AKI) in hospitalized COVID-19 patients. While so far there have been several studies on AKI associated with COVID-19, the present retrospective cohort of patients admitted to two hospitals in United Kingdom during the COVID-19 pandemic offers an additional important contribution to the current literature. This because the relatively large number of patients studied, a suitable study design allowing to properly assess the primary and secondary outcomes, including risk factors which predict AKI as well as the impact of AKI on survival. Notably, the authors have investigated possible unique feature of AKI due to COVID-19 by also studying patients with AKI from any cause and compare the clinical features and outcomes of those with and without COVID-19, a further value of the manuscript Although valuable, however, the study presents several shortcomings that should be addressed to support the conclusions. To mention few of them, i) unclear how baseline serum creatinine for AKI diagnosis was identified; ii) need to discuss the marked discrepancy between mechanical ventilation and development of AKI between the present and other recently published studies; iii) unclear the definition of chronic kidney disease (CKD) in this study; iv) unclear whether patients on ACEi/ARB treatment continued this therapy throughout their hospitalization; v) consider the important suggestions of the statistician (Reviewer 3).

Therefore, I would like to see a revised version of this manuscript.

Abstract

Background- please revise “acute hospitals”

Methods and findings- please provide participant demographics

The last sentence of the methods and findings section must state 2-3 limitations of your work

Author summary

Throughout- please place references in square brackets and format the bibliography in Vancouver style. Please note the full stop must be placed after the square brackets. 

Methods

KDIGO should be introduced on first view on page 5

Please mention details of ethics approval and consent earlier in the methods section

Did your study have a prospective protocol or analysis plan? Please state this (either way) early in the Methods section.

Results

In the section “Factors associated with mortality in COVID-19” there seems to be an = sign missing in the third sentence and a p missing after “care home residents”. There are similar instances on Page 12,13

Please consistently provide p values of up to 3 decimal places, ensuring exact p values are provided unless p<0.001

Discussion

Could you revise “Asians” to “of Asian ethnicity” ? 

Thank you for providing the STROBE checklist. Could I ask you to please provide this as a separate supplementary file and name it S1 Checklist. We also ask that the supplementary information is provided in individual files and referenced in text.

Comments from the reviewers:

Reviewer #1: This is a retrospective cohort study of Acute Kidney Injury associated with COVID-19. There have been several reported studies on Acute Kidney Injury associated with COVID-19 and I am not sure this study provide any novelty given it is already known that high incidence of AKI in patients with COVID-19 that was associated with a higher odds of death.

At least 6 hospitalized patients and 2 ICU cohorts have been reported, in which the investigators have not discussed all of them (http://www.nephjc.com/news/covidaki Sources: Cheng et al KI; Zhou et al, Lancet; Ruan et al, Int Care Med; Chen et al BMJ; Hirsch et al, Kidney Int; Chan et al, MedRXiv; ICNARC dataset (accessed May 8); Mohamed et al, Kidney360)

How to identify baseline serum creatinine for AKI diagnosis has not been described. Different baseline creatinine used may result in different AKI incidence. 

Data on urinalysis and causes of AKI associated COVID have not been described.

If the investigators are able to provide data on hydroxychloroquine, ACEI/ARBs, remdesivir, and safety of their use or as protective/risk factors of AKI, this study can be novel.

Reviewer #2: This is an interesting study in the setting of the COVID19 pandemic and recent recognition of associated severe kidney failure. With all of the recent nephrology-related studies coming out with COVID 19, this study adds to the literature by comparing patients with AKI-associated COVID19+ versus non-AKI COVID19+ patients and AKI-associated COVID19+ versus AKI COVID19- patients. The authors also provide a good comparison between this study's results to other previously published study results conducted in other locations. One question I would ask for these authors is that a recent study by Hirsch et al. published in Kidney International on 7/1/2020 showed that 89.7% of patients on mechanical ventilation developed AKI compared to 21.7% of non-ventilated patients. While this study showed that mechanical ventilation is a similar risk factor for development of AKI, this seems to be a stark discrepancy between mechanical ventilation and development of AKI between the two studies. Any thoughts?

Table 1: It is interesting to note that age, age group, CCF, CKD, and length of hospital stay suggests similar risk factors between AKI and non-AKI patients, regardless of COVID positivity. We do not have "C compared to A", so this statement may not be accurate. It is also interesting that the mortality rate of COVID19 without AKI is almost equivalent to the risk of non-COVID19 patients developing AKI.

Page 12-13: there appears to be several errors in the "Comparison of AKI in patients with COVID-19 and non-COVID disease" section regarding the data in the manuscript versus what is recorded in Table 1. For instance, longer length of stay should be 9.3 vs 8.0 with p=0.76, so the statement that patients with COVID-19 AKI had longer length of stay compared to AKI controls is inaccurate. Other errors included the manuscript's reported gender (should be p=0.04, not 0.003), p-evluaes for white ethnicity, and mechanical ventilation. Would ask the authors to check which is accurate, either Table 1 or what is reported in the manuscript. 

What modalities of RRT was used in your study (e.g. CRRT, iHD, SLED, urgent start PD)?

It was noted that CKD was not associated with increased risk of mortality in patients with AKI, which was a bit surprising. What constituted the definition of CKD in this study? It would be interesting to see in a future study to ascertain how many of the patients that developed need for renal replacement therapy had recovery of renal function or developed CKD/ESRD. I believe the authors appropriately commented on this as well in the discussion section of the study (page 18).

Page 14, Discussion section: "AKI was independently associated with three-fold higher odds of death." I did not see the 3-fold odds ratio, could you clarify on Table 4? I see it was 3 fold higher for the specific population of patients age 65, but that is not what your sentence is saying. Similarly, AKI-associated with COVID-19 was independently associated with an almost four-fold higher odds of death than AKI associated with other AKI -acute illnesses. Can you show the data that shows this, as I didn't see the data presented in Table 1 (mortality 60.5% vs 27.6%?). Forgive me if I overlooked this.

Page 17, your discussion notes that you found that use of an ACEI or ARB was associated with a protective effect with lower odds of death. However, I would question the validity of the statement unless you can ascertain whether patients were continued on ACEi/ARB therapy throughout their hospitalization (in the U.S., many of us stop ACEi/ARB if patients develop severe AKI).

Reviewer #3: I confine my remarks to statistical aspects of this paper. The general approach is fine but I do have some issues to resolve before I can recommend publication.

p. 6 2nd para. Length of stay doesn't seem to have been analyzed

p. 7 1st full para - Why weren't missing data imputed?

 Good for including variables for biological reasons

p. 10 Were any variables included for biological reasons? 

 Don't categorize age. In *Regression Modeling Strategies* Frank Harrell lists 11 problems with this and sums up "nothing could be more disastrous". Leave age continuous and investigate nonlinearies with a spline

 Was colinearity investgated?

Peter Flom

[LINK]

---

## [Decision Letter · Decision Letter 2]

12 Aug 2020

Dear Dr. Kolhe,

Thank you very much for re-submitting your manuscript "Acute Kidney Injury associated with COVID-19: A retrospective cohort study" (PMEDICINE-D-20-02816R2) for review by PLOS Medicine.

I have discussed the paper with my colleagues and the academic editor and it was also seen again by all reviewers. I am pleased to say that provided the remaining editorial and production issues are dealt with we are planning to accept the paper for publication in the journal.

[LINK]

We look forward to receiving the revised manuscript by Aug 19 2020 11:59PM. 

Sincerely,

Adya Misra, PhD

Senior Editor 

PLOS Medicine

plosmedicine.org

Requests from Editors:

Abstract

Please briefly mention where the hospitals are located

Author summary

Could we please tone down “COVID-19 frequently causes AKI” and revise to “Some patients with COVID-19 developed AKI” as causality cannot be inferred from your study design

Introduction

Line 115- please revise “predict AKI” to “associated with AKI” as the study design does not allow predictive diagnoses

Discussion

You may wish to remove “single centre” as the data have been previously reported to be from two hospitals. 

STROBE- please use paragraph and sections as page numbers are likely to change

Comments from Reviewers:

Reviewer #1: The investigators have addressed my concerns; otherwise they have listed in the limitations as appropriated. 

Reviewer #2: Thank you for the answers to my suggested revisions. No further suggested revisions. My prior points to it being an article that does add to the literature remains the same, although there are significant limitations with the use of an observational study performed in a specific demographic population using diagnostic codes and being unable to provide relevant details such as medication use, baseline creatinine values, etc. However these liimitations are noted already by the authors.

Reviewer #3: The authors have addressed my concerns and I now recommend publication.

Peter Flom

[LINK]

---

## [Editor Report · Decision Letter 3]

29 Sep 2020

Dear Dr Kolhe, 

On behalf of my colleagues and the academic editor, Dr. Giuseppe Remuzzi, I am delighted to inform you that your manuscript entitled "Acute Kidney Injury associated with COVID-19: A retrospective cohort study" (PMEDICINE-D-20-02816R3) has been accepted for publication in PLOS Medicine. 

PRODUCTION PROCESS

Before publication you will see the copyedited word document (within 5 busines days) and a PDF proof shortly after that. The copyeditor will be in touch shortly before sending you the copyedited Word document. We will make some revisions at copyediting stage to conform to our general style, and for clarification. When you receive this version you should check and revise it very carefully, including figures, tables, references, and supporting information, because corrections at the next stage (proofs) will be strictly limited to (1) errors in author names or affiliations, (2) errors of scientific fact that would cause misunderstandings to readers, and (3) printer's (introduced) errors. Please return the copyedited file within 2 business days in order to ensure timely delivery of the PDF proof. 

If you are likely to be away when either this document or the proof is sent, please ensure we have contact information of a second person, as we will need you to respond quickly at each point. Given the disruptions resulting from the ongoing COVID-19 pandemic, there may be delays in the production process. We apologise in advance for any inconvenience caused and will do our best to minimize impact as far as possible.

PRESS

PROFILE INFORMATION

Thank you again for submitting the manuscript to PLOS Medicine. We look forward to publishing it. 

Best wishes, 

Adya Misra, PhD

Senior Editor 

PLOS Medicine

plosmedicine.org